# Machine learning in top quark physics at ATLAS and CMS

**Matthias Komm[1⋆], for the ATLAS and CMS collaborations**

**1** DESY, Hamburg, Germany

⋆ matthias.komm@cern.ch

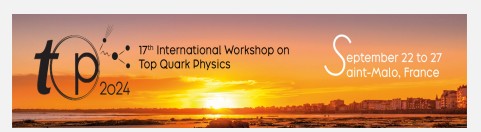

## Abstract

**This note presents an overview of machine-learning-based techniques used in the study of the top quark. The research community has developed a diverse set of ideas and tools, including algorithms for the efficient reconstruction of recorded collision events and innovative methods for statistical inference. Recent applications by the ATLAS and CMS collaborations are also highlighted.**

## 1 Introduction

In recent years, machine learning (ML) and artificial intelligence have revolutionized numerous fields. In top quark research, ML has already been a driving force for over a decade. Given the top quark's nearly exclusive decay to a b quark and a W boson, the continuous advancements in b jet identification through increasingly sophisticated ML algorithms is a prime example [1,2]. Stretching from the discovery of single top quark production at the Tevatron [3,4] to the recent observation of four-top-quark events by the ATLAS [5,6] and CMS [7,8] collaborations at the CERN LHC highlight the critical role of ML in achieving such milestones in the field.

This note reviews a collection of state-of-the-art ML algorithms addressing various aspects of top quark research, including top quark and event reconstruction, analysis strategies, and novel methods for statistical inference. An outlook on the impact of ML on future research at the upcoming HL-LHC is also provided.

## 2 Top quark reconstruction

The reconstruction of top quark events typically involves two stages. First, the inference of the direction(s) of undetected neutrino(s) from leptonically decaying top quarks, $t \rightarrow b\ell\nu$, if present. Second, the association of the remaining decay products (leptons and jets) to each top quark in the event. Various ML algorithms have been developed to address both tasks.

The $\nu$-FLOW approach [9,10] utilizes a normalizing flow neural network (NN) conditioned on reconstructed event observables. This network maps the true neutrino direction vector, derived from the event generator truth, to a three-dimensional normal distribution. By sampling from this distribution, the likelihood of possible neutrino directions can be inferred from data, as illustrated in Fig. 1a for a representative event, demonstrating superior performance over regressing the values via a feed-forward NN or employing a W boson mass constraint.

An efficient solution for assigning all top decay products to reconstructed particles is provided by the SPANET approach [11]. This method employs an NN transformer architecture with over 10M parameters. The latest version of SPANET also incorporates auxiliary targets, such as the regression of the neutrino direction (as shown in Fig. 1b) and the discrimination of signal from background events.

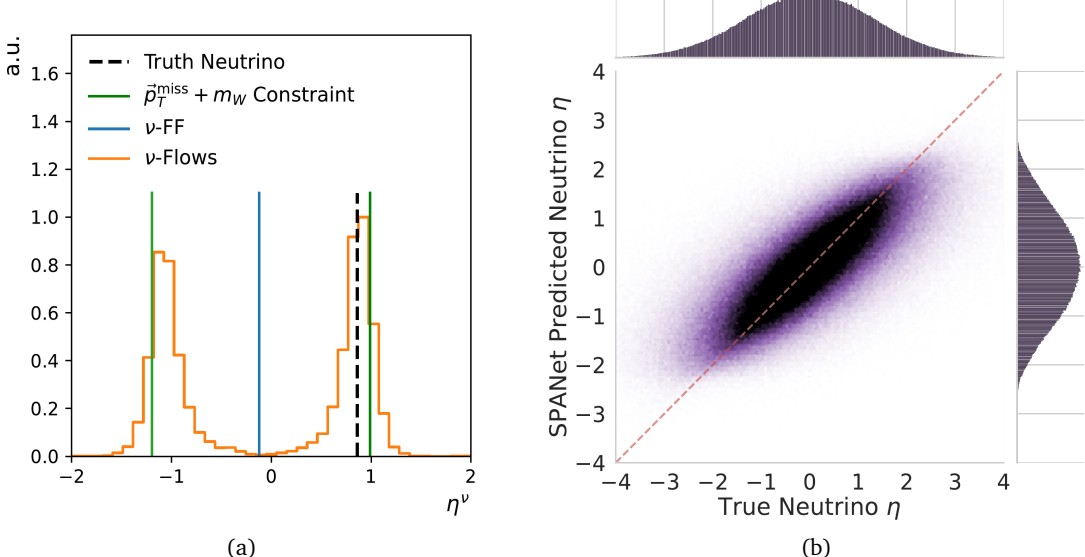

(a)

(b)

Figure 1: Inference of the pseudorapidity of the neutrino originating from a leptonically decaying top quark: (a) $\nu$-FLOW approach [9]; (b) SPANET approach [11].

A novel approach to associate the decay products to reconstructed particles is given by the HYPER method [12]. In this framework, the top quark decay products are represented as hypergraphs—a generalization of graph NNs, where each edge can connect more than two nodes. Despite using only 345k parameters, the NN achieves a performance comparable to SPANET.

## 3 Analysis strategy

A crucial component of many physics analyses is determining the rate of background events. This is particularly challenging for analyses focusing on multijet events, a common feature of top quark analyses, which have to deal with a non-negligible fraction of events arising from pure QCD interactions. The precise yield and distribution of these interactions are difficult to predict accurately through simulation.

One solution is the so-called ABCD matrix method, where the background fraction is extrapolated using data from orthogonal regions defined by two independent observables. The DISCO method [13] introduces an NN classifier to construct suitable observables that are uncorrelated and simultaneously separate the signal from background processes. This is achieved

by introducing an additional penalty term during the training to suppress distance correlations either between the scores of two NN classifiers or between one score and an auxiliary observable.

Another approach was showcased in a recent analysis by the CMS collaboration searching for all-hadronic four-top-quark events [14]. An auto-regressive normalizing flow was employed to transform data events from a background-enriched region into the signal region.

# 4 Statistical inference

In addition to improving event reconstruction and analysis strategies, new ML-based tools have been studied to enhance the statistical inference, going beyond the traditional (binned)-likelihood approach.

Likelihood-free inference, also known as simulation-based inference, focuses on using the output score $s$ of a classifier as a test statistic directly. This approach exploits the fact that $s$ can provide a likelihood ratio $H_1/H_0 = s/(1-s)$, which is essential for hypothesis testing, ie. rejecting the null hypothesis $H_0$ in favour of an alternative hypothesis $H_1$. Examples of tools employing this method include INFERNO [15] and SALLY [16], both of which can account for sources of systematic uncertainties as well.

Another domain where ML proves valuable is unfolding of differential cross sections, enabling direct comparisons with theoretical predictions or results from other experiments. However, it is crucial to note that no ML algorithm can overcome the inherent ill-posed nature of the unfolding problem (i.e., inverting a Fredholm integral equation), necessitating some form of regularization. The OMNIFOLD approach [17] facilitates unbinned and multidimensional unfolding through an iterative procedure in which differences between simulation and data are learned via a classifier and subsequently reweighted to match the distributions in the simulated signal sample to data. The number of iterations controls the degree of regularization.

This method has been successfully demonstrated by the ATLAS and CMS collaborations for unfolding Drell-Yan [18] and minimum bias events [19], respectively. Selected results are shown in Fig. 2. Notably, the unbinned nature of OMNIFOLD enables the unfolding of novel quantities, such as taking the average of the jet mass as a function of another observable.

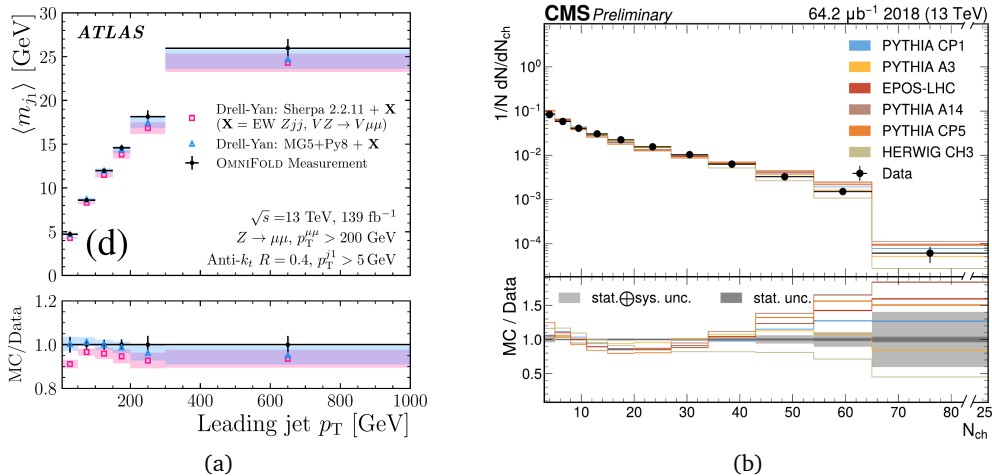

Figure 2: Examples of unbinned, multidimensional unfolding with OMNIFOLD [17]: (a) average jet mass as a function of the jet $p_T$ in Drell-Yan events [18] by ATLAS; (b) number of charged jet constituents in minimum bias events [19] by CMS.

## 5 HL LHC outlook

The upcoming high-luminosity (HL) phase of the LHC will significantly increase computing demands for analyzing the much larger datasets. An actively researched area is the use of ML to reduce the reliance on simulated samples or enhance the speed of detector simulations.

The CMS collaboration has recently addressed the former challenge by introducing the DCTR method [20, 21] for reweighting simulated samples. The study comprised the reweighting of samples to emulate parameter shifts used for estimating the impact by systematic uncertainties. An example variation involving the $h_{\mathrm{damp}}$ parameter of the POWHEG event generator [22] is shown in Fig. 3a. The good agreement observed after reweighting demonstrates that this technique could replace the need for generating dedicated samples in the future.

Another application of the DCTR method is reweighting samples to achieve higher-order accuracy. In Fig. 3b, an NLO sample has been successfully reweighted to match an NNLO prediction.

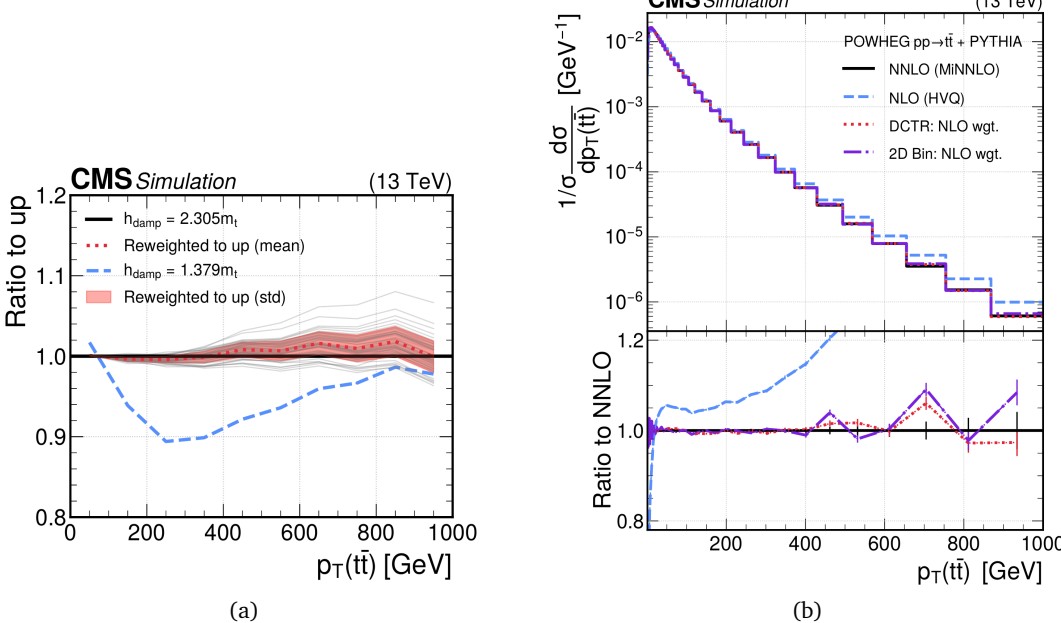

(a)                                                    (b)

Figure 3: Reweighting of simulated events using the DCTR method [20] by CMS: (a) reweight to emulate a variation of the $h_{\mathrm{damp}}$ parameter of the POWHEG event generator; (b) reweight to emulate NNLO accuracy from an NLO sample. Figures are taken from Ref. [21].

## 6 Conclusion

For over a decade, ML has been a driving force in top quark research. Novel ML-based approaches in top quark reconstruction, background estimation techniques, and innovative tools for statistical inference are also setting the stage for the upcoming high-precision era of the HL LHC.

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
