# Peer review of "Machine learning in top quark physics at ATLAS and CMS"

_SciPost Physics Proceedings_

## Round 1 · Referee Report · Samuel Calvet (Referee 1) · 2025-5-5

Strengths
1- The proceedings is clearly written and the presentation of ideas is easy to follow, making the review accessible to a broad audience. 2- The paper provides a well-structured and comprehensive overview of ML-based techniques that are or could be used for top physics
Weaknesses
1- The title and the abstract presents the proceedings as a review of ML techniques used in top physics. Actually only a small fraction of these techniques are used in this field. The rest of these technique are not (yet) used. 2- In section 5, the improvement brought by the ML is not clear. What is the difference with regular reweighting techniques, for example base on matrix element of kinematic ?
Report
I would encourage the authors to address the 2 identified weakness.
Requested changes
1- Some sentences should be rephrased to clarify that certain techniques discussed—though not currently standard in top quark physics—could offer valuable contributions to the field if appropriately adapted and adopted. 2- Section5 should more explicitly articulate the advantages introduced by the machine learning approach, particularly in contrast to classical methods, to better highlight the added value of the proposed strategy. 3- page 2, 1srt paragraph: "illustrating superior" would be more precise than "demonstrating superior" 4- Figure 1: caption could be clearer if you specify that is for semi-leptonic ttbar events
Recommendation
Ask for minor revision

---

## Editorial Decision

unknown